# Cellulose Hydrogel with Hyaluronic Acid and Silver Nanoparticles: Sustained-Release Formulation with Antibacterial Properties against *Pseudomonas aeruginosa*

**DOI:** 10.3390/antibiotics12050873

**Published:** 2023-05-08

**Authors:** Mirian Sumini, Clara Ruiz de Souza, Gabriel Jonathan Sousa Andrade, Igor Roberto Cabral Oliveira, Sara Scandorieiro, Cesar Augusto Tischer, Renata Katsuko Takayama Kobayashi, Gerson Nakazato

**Affiliations:** 1Department of Microbiology, Biological Sciences Center, Londrina State University, Londrina 86057-970, Paraná, Brazil; 2Department of Pharmacy, Health Sciences Center, Londrina State University, Londrina 86057-970, Paraná, Brazil; 3Department of Biochemistry and Biotechnology, Exact Sciences Center, Londrina State University, Londrina 86057-970, Paraná, Brazil; 4Department of Civil Engineering, Faculty of Technology, Federal University of Amazonas, Manaus 69077-000, Amazonas, Brazil; igoroliveira@ufam.edu.br

**Keywords:** hydrogel topic, nanocomposite, silver nanoparticles, pathogenic bacteria, antibacterial properties

## Abstract

Pathogenic bacteria resistant to conventional antibiotics represent a global challenge and justify the need for new antimicrobials capable of combating bacterial multidrug resistance. This study describes the development of a topical hydrogel in a formulation composed of cellulose, hyaluronic acid (HA), and silver nanoparticles (AgNPs) against strains of *Pseudomonas aeruginosa*. AgNPs as an antimicrobial agent were synthesized by a new method based on green chemistry, using arginine as a reducing agent and potassium hydroxide as a carrier. Scanning electron microscopy showed the formation of a composite between cellulose and HA in a three-dimensional network of cellulose fibrils, with thickening of the fibrils and filling of spaces by HA with the presence of pores. Ultraviolet-visible spectroscopy (UV-vis) and particle size distribution for dynamic light scattering (DLS) confirmed the formation of AgNPs with peak absorption at ~430 nm and 57.88 nm. AgNPs dispersion showed a minimum inhibitory concentration (MIC) of 1.5 µg/mL. The time–kill assay showed that after 3 h of exposure to the hydrogel containing AgNPs, there were no viable cells, corresponding to a bactericidal efficacy of 99.999% in the 95% confidence level. We obtained a hydrogel that is easy to apply, with sustained release and bactericidal properties against strains of *Pseudomonas aeruginosa* at low concentrations of the agent.

## 1. Introduction

Bacterial resistance to conventional antibiotics has become a global public health problem. The World Health Organization (WHO) published in 2017 to guide and promote research and development of new antimicrobials, a list of “priority pathogens” resistant to antibiotics and which represent the greatest threat to human health [1].

*Pseudomonas aeruginosa* were included in the critical group on the WHO list, as they are Gram-negative bacteria multiresistant to antibiotics, including carbapenems and third-generation cephalosporins, currently the best antibiotics available for treatment [1]. *P. aeruginosa* has become a pathogen capable of causing skin and soft tissue infections through the formation of biofilms and subsequent chronic wounds, with built-in abilities to find new ways to resist treatment and exchange genetic material, transferring antibiotic resistance to other bacteria [2].

Cellulose has been highlighted as an attractive, sustainable, non-toxic, and renewable material for water-based gels (hydrogels). Although various cellulose derivatives are widely used in drug delivery systems, their application with improved performance, such as sustained release, has only emerged in recent years [3,4]. Hydrogels with antibacterial properties have been shown to minimize bacterial colonization during wound healing and reduce the risk of infection [5,6,7].

The hydrogels are matrices formed by a porous three-dimensional network of crosslinked polymer chains, helix-like macromolecules, or colloid particles that retain large amounts of water (up to 99.9%) [8,9]. The hydrogels derived from cellulose are characterized by remarkable biocompatibility, viscoelasticity, controllable porosity, chemical adaptability, and unique pharmacokinetic behavior [10,11,12]. In addition, they differ from other polymers due to their constitutive polymers being insoluble in water by forming colloidal suspensions [3,12]. The size of the constitutive unit of the biopolymer is also larger compared to other polymers, with fibril chains with diameters of nanometers and a length of up to microns [13,14], which allows the formation of a network with suitable electrostatic, thermal, and mechanical properties [15], providing high stability to the gel structure [12,16].

At the application level, the importance of cellulose-derived hydrogels lies in the ability of the hydrogels to mimic the physical and biochemical conditions found in the human body or required by biological systems, including an inherent similarity to collagen fibrils [17,18,19,20,21]. Cellulose primarily affects the properties of the hydrogel in terms of morphology, structure, and crystallinity. Moreover, due to its chemical bonds, created with the other ingredients of the formulation, it presents better physicochemical properties [12,22,23].

In hydrogels, the release of an active is caused by diffusion and is related to the physical–chemical interactions of its constituents, as well as the swelling behaviors of the gel [24], in which more significant swelling leads to higher release rates [25]. Moreover, the swelling capacity allows the control of pore size [26], an essential factor in the support and release of particles such as cells, proteins, and nanoparticles [8].

The antibacterial activity of metallic compounds, mainly silver nanoparticles (AgNPs), seems to be related to the release of silver atoms or ions [27,28], as well as the size and morphology of the nanoparticles (round, rod, triangle, etc.) [10,11,13,29,30,31,32]. Ag ions can interact with cell membranes, nucleic acids, and bacterial proteins, leading to blockage of metabolic pathways such as respiratory chains, transport mechanisms, DNA denaturation, and leakage of bacteria contents [33]. AgNPs can also interact with bacterial surfaces due to the high affinity of Ag ions and sulfur- and phosphorus-containing compounds on the bacterial cell walls, which facilitates the binding of AgNPs to the bacterial surface and entry into bacterial cells, ultimately leading to lysis [28].

The toxicity of AgNPs in humans and the environment has been intensely investigated [8,16,18,34]. However, studies indicate that the broad spectrum of antimicrobial action of AgNPs occurs at low concentrations, which limits or reduces toxicity to the cells of mammals [35,36,37,38].

It is known that the stabilizing agents of AgNPs play a vital role in their toxicity and antibacterial activity [28,39]. The reducing agent and stabilizer control the process and thus hold the size and shape of the AgNPs [40]. Sodium citrate, sodium borohydride, dimethylformamide, triethanolamine, and hydrazine can be used as electron donors for the reducing process and production of nanoparticles [41,42]. It is noteworthy that some of these reducing agents are harmful. Therefore, the search for substitutes, such as more sustainable green chemistry, is the goal of many researchers [13,17]. Previously, Ilyin et al. 2011 [43] obtained a hydrogel containing AgNPs using the amino acid cysteine as a reducing agent.

Biologically active amino acids are environmentally friendly compounds suitable for green chemistry. Arginine presents inorganic interaction with metals, acting as a Lewis base with lone pairs of electrons in the amine and carboxyl groups, allowing the connection with unoccupied transition metal orbitals, producing the metallic complex [44,45,46]. The metallic complexes given by Arg/Ag^+^ are remarkably strong, with high binding energy [41], potentially leading to a molecular structure with suitable shelf life, bactericidal, biocompatible, and very low environmental impact [47,48,49].

In this study, we developed a biocompatible hydrogel for topical use composed of cellulose with HA and incorporated with AgNPs as an antibacterial active against pathogenic strains of *P. aeruginosa*. The hydrogel formulation constituted a system for sustained and targeted release of the active compound, which was stable, easy to apply, and displayed high bactericidal efficiency.

## 2. Results

### 2.1. Final Formulation of the Cellulose Hydrogel

The formulation (Figure 1) was composed of nanocellulose combined with hyaluronic acid gels as the technological base, enriched with AgNPs as the active ingredient, potassium sorbate as a preservative, and purified water as the vehicle. The topical hydrogel composition presented key properties for a potential skin infection treatment, such as a well-integrated base.

### 2.2. Characterization of the Composite Consisting of Cellulose and Hyaluronic Acid (HA)

#### 2.2.1. Scanning Electron Microscopy (SEM)

The morphology of the cellulose fibrils and the combination of cellulose and HA were investigated by SEM (Figure 2). The results of cellulose micrographs (Figure 2A,B) showed fibrils in a random arrangement and three-dimensional structure with the presence of small pores. The micrographs of the composite formed from cellulose with HA (Figure 2C,D) showed thicker and more compact fibrils, constituting a dense network in several layers with varied lengths and perception of depth reduction. The fibrils’ average length and standard deviation measured were 15.3 ± 3.5 µm. The mean diameter of cellulose and cellulose fibrils with HA were 103.67 and 107.6 nm, respectively, resulting in the aspect ratio range (length/width) between 63.5 and 71.8.

#### 2.2.2. Characterization by FT-IR

Analysis by Fourier transform infrared spectroscopy (FT-IR) (Figure 3) showed the presence and integrity of the biopolymers and their chemical influences, which were accompanied by the change of characteristic bands. The incorporation of HA into cellulose fibrils (Figure 3B) was verified by the appearance of a wider band exactly at 3454 cm^−1^ when compared to the absorption band characteristic of cellulose (band of 3294 cm^−1^) (Figure 3A) and which can be attributed to the stretching of the bonds (OH and NH). The band at 2368 cm^−1^ was related to the (CH) vibration, while the 1627 cm^−1^ band was to the amide (C=O) groups and intermolecular water. At 1109 cm^−1^, there were vibrations related to (CO), and at 678 cm^−1^, out-of-plane deformation of (C-OH).

#### 2.2.3. Characterization by Differential Scanning Calorimetry (DSC)

The thermal degradation for the cellulose gel and for the cellulose gel with HA is represented in Figure 4. The cellulose gel (Figure 4A) showed the occurrence of an endothermic event at 67 °C and two exothermic events at 359 °C and 420 °C. For the cellulose gel with HA (Figure 4B), there was one endothermic event at 62 °C and two exothermic events at 217 °C and 335 °C.

For both samples (Figure 4A,B), the endothermic events are related to the degradation of low molar mass compounds present in the HA molecule, such as the loss of water distribution. The occurrence of the first exothermic event at 359 °C (Figure 4A) and 217 °C (Figure 4B) is related to the sample mass loss due to bond breaking, leading to the formation of CO_2_, water, and other substances derived from hydrocarbons. The third event, 335 °C (Figure 4A) and 420 °C (Figure 4B), shows the continuous sample mass loss.

The DSC curve showed a single mass loss process for the cellulose gel with HA (Figure 4B), characterizing the interaction between the two biopolymers and the thermal stability effect for the composite formed between cellulose and HA.

### 2.3. Synthesis and Characterization of AgNPs

The titration and reduction reaction temperature were tested over a temperature range of 10 °C to 90 °C. A color change from transparent to dark orange was observed only at the maximum temperature of 90 °C. UV-vis analysis of the solution revealed an absorption spectrum with a peak at 430 nm (Figure 5), indicating the formation of silver nanoparticles. PSD measurements further confirmed the formation of AgNPs with an average size of 57.88 nm.

### 2.4. Antibacterial Activity Assays

#### 2.4.1. Disk Diffusion on Agar Test

The disk-diffusion test has shown that the synthetic process with alkaline arginine succeeds in synthesizing AgNPs (Figure 6). The presence of an inhibition halo with a diameter of 8 mm around the disk previously embedded with AgNPs indicates the antibacterial activity of these nanometals against *Pseudomonas aeruginosa*.

#### 2.4.2. Determination of Minimum Inhibitory Concentration (MIC)

The MIC for AgNPs was 1.5 µg/mL, which can be considered highly effective against *P. aeruginosa*.

#### 2.4.3. Time–Kill Assay

The time–kill curve (Figure 7) showed that after a 3-hexposure to the hydrogel containing AgNPs, there were no viable cells, corresponding to a bactericidal efficacy of 99.999%. The bactericidal activity remained throughout the experiment, which indicates that AgNPs have a prolonged effect.

#### 2.4.4. Data Analysis Statistical between Minimum Inhibitory Concentration (MIC) and Time–Kill

In the relation of the minimum inhibitory concentration (MIC) with the time–kill, unifactorial Anova showed that there is a significant decrease in the means of the concentrations [F (7, 16) = 4.831; *p* = 0.00435]. Through the post hoc test, Tukey HSD identified that the significant difference is between the control dilutions and GEL-AgNP in the time from the 3 h of analysis, but not between the other dilutions and their respective times.

### 2.5. Cytotoxicity Assay with Human Red Blood Cells (RBC)

AgNPs dispersion was non-toxic to RBC at all concentrations tested. Hemolytic activity was not observed even at the highest concentration tested (5500 mg/mL), which corresponded to only 1.1% of cell death (Figure 8).

## 3. Discussion

Antibiotic drugs are the basis for the treatment of bacterial infections. However, their overuse has led to an increase in multidrug-resistant bacteria, making drugs ineffective. Resistance to antimicrobials is one of the biggest global public health problems, causing limitations in therapies. Furthermore, almost all antimicrobial substances at high concentrations have adverse effects on the host and the environment, thereby increasing the search for new compounds that can inhibit multidrug resistance.

*P. aeruginosa* is widely known for its association with high infection rates, its variety of intrinsic resistance mechanisms, biofilm production, and plasmid-encoded antimicrobial resistance genes [49]. This species acquires additional mechanisms of resistance through mutations and the acquisition of mobile genetic elements, such as the horizontal transfer of genes that encode beta-lactamases (penicillinases, cephalosporinases, and metallo-beta-lactamases), as well as efflux pumps [50,51,52].

It is well known that AgNPs have a broad spectrum of bactericidal activity by binding to the cell membrane and increasing its permeability due to structural changes that promote cell lysis. AgNPs induce membrane rupture, creating gaps for the entry of Ag ions into the cell [53].

The search for processes based on green and sustainable chemistry has driven this work. The present study involved the development of a sustained-release hydrogel, a topical formulation composed of cellulose, HA, potassium sorbate, and AgNPs, presenting a formulation with high bactericidal action. AgNPs were produced by the reduction of silver ions in solution using AgNO_3_ as a precursor agent and the amino acid arginine as a reducing agent associated with potassium hydroxide as a carrier.

The proposed synthesis was successful when dark or reddish-orange was achieved. As described earlier, the characteristic absorption peak at ~430 nm can be related to particles of 50–60 nm, a well-covered correlation between the plasmonic absorption peak and particle size [54]. The shape and size of the nanoparticles were the main factors that raised the spectral range of resonance and the corresponding wavelength, as well as the effective mass, electronic density, and interaction with stabilizing agents. The larger the particle, the longer the optical absorption wavelengths [55].

The UV-vis spectrum of AgNPs synthesized with arginine: KOH: AgNO_3_ at 90 °C exhibited a single plasmonic surface resonance band, indicative of the spheroid shape and the absence of anisotropic particles. The wide range of wavelengths, the center of the absorption peak, and its intensity are correlated with size; surface coverage rate and size are inversely correlated. At higher surface coverage rates, long-range order appears with the formation of ordered structures. Typically, the wavelength resonance at 400 to 450 nm presents values close to ~27% for spheroid structures ~50 nm wide [56,57].

Arginine presents inorganic interaction with metals, acting as a Lewis base due to the pairs of free electrons in the amine and carboxyl groups, allowing it to bind with unfilled orbitals of transition metals, originating metallic complexes. This metallic process has high-affinity values, and its binding enthalpy in the neutral form is −80.5 kcal.mol^−1^ at 298 k, and free energy of −70.3 kcal.mol^−1^, both remarkable numbers [41]. Shoeib et al. [48] showed that the bond is formed by the interaction of the terminal nitrogen, the second with the carboxylic oxygen, and the amide nitrogen. Thus, the interaction between arginine and the Ag ions establishes a molecular, bactericidal, time-stable structure with low inherent toxicity and reduced risk of environmental impact.

The antibacterial effect of AgNPs is dependent on particle size, with smaller diameters exhibiting higher activity [10]. Choi et al. [58] have correlated the dimensions of AgNPs and their permeability to bacterial membranes, with smaller particles being more effective as antibacterial agents. AgNPs within the size range of 20–80 nm have been reported to have antimicrobial activity [59,60].

Analysis of the present hydrogel by electron micrographs obtained by SEM showed the morphological characteristics of the cellulose fibrils (without the addition of HA) and the composite between cellulose and HA. The images show that the pure cellulose fibrils formed a random arrangement in a three-dimensional structure with the presence of pores. The combination of cellulose with HA resulted in the filling of the cellulose fibrils by HA, with thicker fibrils and the presence of pores, confirming that the addition of HA provided significant differences in the micromorphology of the composite.

Cellulose usually has a three-dimensional structure with small pores, which tend to fill when blended with HA [61,62]. As cellulose is primarily composed of high crystalline fibrils that are several nanometers in length but short in diameter, these fibrils interact at contact points, creating niches filled with solvent and dissolved substances. These niches act as containers that inter-communicate, creating pores for solvent and solute dispersion.

The non-covalent interactions between the constituents of the hydrogel and the cellulose are vital to prolonging the release of the active agent as a carrier for sustained administration. It is a result of the cellulose tridimensional ordering that holds HA in its scaffolding structure.

The absorption and release features of cellulose hydrogels are critical factors influencing the release rates, especially during the initial stage (within the first 12 h). This can be achieved by modulating the distances between the chains network, which act as pores, and being primarily composed of water, easily allowing drugs to diffuse through the hydrogel [12]. Therefore, the addition of HA provided an efficient strategy to reduce the pore sizes in the cellulose hydrogel.

The FT-IR analysis revealed the presence and integrity of the biopolymers, which was confirmed by the effect of the chemical composition, accompanied by the change of characteristic bands. The spectra of cellulosic materials normally show an absorption band in the range of 3350 cm^−1^ [63,64,65]. The incorporation of HA into the nanocellulose network can be validated by the presence of a wide band at 3454 cm^−1^, revealing intermolecular interactions between cellulose and HA via OH/NH groups. Thus, the addition of HA in its non-ionized form allows interactions with cellulose fibrils to occur via NH groups. The observed absorption is greater than the characteristic absorption band of cellulose described in the literature [63,64], as well as the absorption obtained from our analyses for pure nanocellulose at 3294 cm^−1^.

Comparatively, Haxaire et al. [66] studied the carboxylate groups of HA spectra with dry hyaluronic acid hybrid films, replacing H^+^ with Na^+^ and identified that the CO carboxyl band occurs at 1745 cm^−1^ and the COOH band with a maximum at 1220 cm^−1^. The authors concluded that the band around 1745 cm^−1^ does not appear in HA but occurs in hybrid films, evidencing that HA is in the sodium form, while in the hybrid form, it is protonated.

The physical properties of thermal stability based on temperature variation were obtained by DSC. The thermal degradation showed characteristics of a hygroscopic sub-stance characterized by one endothermic and two exothermic events. When nanocellulose is homogeneously distributed, it indicates an increase in the surface area, which is caused by a material with a larger crystalline region, related to the exothermic events and the temperatures of 359 °C and 420 °C for pure cellulose and 217 °C and 335 °C for composite between cellulose and AH [67,68].

The endothermic events that occurred at 67 °C for pure cellulose and 62 °C for the composite between cellulose and AH are related to the loss of mass in the sample. The occurrence of peaks showing a single sample mass loss process for both biopolymers demonstrates the interaction and thermal stability of the constituted composite.

Pure cellulose is more thermally stable than the compound between cellulose and AH, as demonstrated by the *T_onset_* temperatures (onset of thermal combustion). The mass loss at lower temperatures may be associated with the availability of HA, which is more easily susceptible to thermal degradation.

The hydrogel developed using silver nanoparticles as an antimicrobial agent was proposed as an alternative for the control of bacterial infections, particularly *P. aeruginosa*, as this bacterium is one of the most common pathogens found in skin wounds and is responsible for the formation of biofilms, which subsequently leads to chronic wounds [2]. Comprising a technological system formed by cellulose with hyaluronic acid with promising extended-release properties could be more efficient than conventional drugs by enabling targeted delivery to the sites of infection in a continuous and controlled manner. The disk diffusion on the agar test showed that the synthetic process with alkaline arginine produced AgNPs with antibacterial activity against *P. aeruginosa*. The disk-diffusion assay demonstrated the antimicrobial activity of the AgNPs dispersion, as evidenced by the formation of an inhibition zone with an approximate diameter of 8 mm around the disk.

The antimicrobial activity around the disk previously soaked with AgNPs dispersion generated an inhibition halo approximately 8 mm in diameter.

The limitation of this test is related to the diffusion of the compound in the medium as an impediment that the agar imposes, as well as the volume of the active [67].

The present study showed that the MIC value of AgNPs against *P. aeruginosa* was 1.5 µg/mL, which indicates that these nanoparticles are highly effective against this bacterial species. Liao et al. [48] reported similar results to ours, with AgNPs MIC ranging from 1.406 to 5.625 µg/mL against *P. aeruginosa*, including multidrug-resistant strains. By incorporating silver nanoparticles into the hydrogel, we obtained an effective formulation for the treatment of bacterial infections at low concentrations of the agent, thereby reducing the toxicity of silver to cells and making it difficult for microbial resistance to develop against the nanoparticles.

Mellot et al. [69], in studies testing the efficacy of topical therapy using conventional antibiotics formulated with 1% silver sulfadiazine and used in the particular case of infections caused by P. aeruginosa, it was shown that the drug provided an eradication of approximately 85%. The time of death assay performed on the final product, a hydrogel containing AgNPs, showed a bactericidal efficacy of 99.999% against *P. aeruginosa* after 3 h of action.

Our RBC toxicity assay revealed that the AgNPs were not cytotoxic to human blood cells. Our metal compound exhibited no cytotoxic effect on RBC at the studied concentrations of 43.0–5500 mg/mL, though it was not possible to calculate the CC50 for human erythrocytes. In our study, the highest concentration tested (5500 mg/mL) resulted in only 1.1% hemolysis. Similarly, Scandorieiro et al. [70] reported a hemolysis rate of only 1.4% at the highest concentration of biogenically obtained silver nanoparticles (bio-AgNP) tested (250 µM).

Arginine is the most alkaline of the 20 biological amino acids, and *P. aeruginosa* is susceptible to the pH changes it causes in the environment, which play a role in bacterial biofilm development and diseases such as cystic fibrosis, in which the lungs have an acidic pH [68]. The use of arginine as a carrier for nanoparticulate silver seems to be a promising therapeutic alternative, as indicated by the results obtained.

The hydrogel formulated with silver nanoparticles showed potent inhibitory activity against the pathogenic strain *P. aeruginosa*, even at low concentrations. The product displayed several advantageous properties, such as a well-integrated base, residual efficacy, ease of application, and high bactericidal activity. The composite system made up of cellulose with HA allowed for the efficient release of AgNPs, which could be targeted to the site of infection in a continuous, sustained, and controlled manner, offering an advantage over conventional drugs.

## 4. Materials and Methods

### 4.1. Bacterial Strains

All antimicrobial assays were performed against reference strains from the American Type Culture Collection (ATCC) provided by the Laboratory of Basic and Applied Bacteriology of Londrina Stated University (Londrina, Paraná, Brazil). The strain *Pseudomonas aeruginosa* (ATCC 9027) was used and stored in 25% glycerol (Merck) at 80 °C.

### 4.2. Synthesis of Silver Nanoparticles (AgNPs)

The silver nitrate (AgNO_3_), potassium hydroxide (KOH), and arginine (Arg) were purchased from Sigma-Aldrich (St. Louis, MO, USA). The AgNPs were prepared by titration of the AgNO_3_ solution against the reducing solution. The concentration of each reagent was 0.0618 mol/L for AgNO_3_ (source of silver ions), 0.1236 mol/L for Arg (reducing agent), and 0.309 mol/L for KOH (carrier). The molar ratio was 1:2:5, [AgNO_3_]: [Arg]: [KOH]. The reducing solution was prepared, dissolving, i.g., 0.4305 g of arginine and 0.3466 g of KOH in 20 mL of deionized water. For the silver ion solution, 0.2610 g of AgNO_3_ was dissolved in 25 mL of deionized water, then stirred at 90 °C. The former reducing solution was titrated with AgNO_3_ solution using a syringe pump at a flow rate of 10 mL/h under agitation at 90 °C. The reaction was considered complete when the color changed to dark orange.

### 4.3. Characterization of Silver Nanoparticles (AgNPs)

#### 4.3.1. UV-Vis Spectrophotometry

The ultraviolet spectra of the AgNPs diluted samples (1:10, *v/v*) were carried out on a Jenway 6705 UV-vis spectrophotometer (Fisher Scientific, Waltham, MA, USA) at room temperature. The spectra were generated by scanning samples on quartz cuvettes ranging from 200 to 600 nm, with 0.1 nm of resolution.

#### 4.3.2. Particle Size Distribution

The particle size (PSD) of AgNPs was analyzed by DLS (Zeta-APS, Matec Applied Sciences, Northborough, MA, USA). Measurements were performed in a concentrated sample (3%), with the 26 °C employing acoustic attenuation (dB/cm) vs. sound frequency (1 to 100 MHz) of colloidal dispersion.

#### 4.3.3. Cytotoxicity Assay with Human Red Blood Cells

The hemolytic activity of AgNPs was determined according to Izumi et al. [71], with modifications. Blood was collected from a healthy human donor with voluntary consent into heparinized tubes (Vacutainer BD, Curitiba, Paraná, Brazil). The collection was approved by the human ethics committee (CAAE 47661115.0.0000.5231, No. 1.268.019—UEL). Erythrocytes were separated by centrifugation (5000 rpm, 4 °C for 5 min) and prepared at 6% (*v/v*) in phosphate-buffered saline (0.1 M PBS and pH 7.2). PBS was composed of 0.9% (*w/v*) sodium chloride (Merck, Darmstadt, Frankfurt, Germany), 0.2 M monobasic sodium phosphate (Chemco, Hortolandia, São Paulo, Brazil), and 0.2 M dibasic sodium phosphate (Dinâmica, São Paulo, São Paulo, Brazil). A volume of 100 µL of 6% human red blood cells (HRBC) and 100 µL of PBS with different concentrations of the AgNPs were added to 96-well microplates (Corning^®^, Somerville, MA, USA). The microplate was incubated for 3 h at 37 °C, and the supernatants were analyzed at 550 nm to monitor the release of hemoglobin. Triton X-100 (Sigma Aldrich, Saint Louis, MO, USA) at 1% was used as a control for 100% hemolytic activity, and the percentage of hemolysis was calculated for each concentration (43.0–5500 mg/mL) of the AgNPs.

### 4.4. Antibacterial Activity Assays for AgNPs

#### 4.4.1. Disk Diffusion on Agar Test

The qualitative analysis of the antibacterial effect of the synthesized AgNPs was observed by disk diffusion in agar Müller–Hinton (BD Difco, Franklin Lakes, NJ, USA). A bacterial saline suspension (0.85% NaCl), with turbidity adjusted to 0.5 McFarland scale corresponding to 1.5 × 10^8^ colony-forming units (CFU)/mL, was inoculated on MHA Petri plates. Sterilized paper disks with 0.01 mL of AgNPs (130 mg/mL) were placed equidistantly on the plates, with two disks per sample. After incubation for 18–24 h at 37 °C in a microbiological oven, the radium of the inhibition zone was measured and registered in millimeters.

#### 4.4.2. Determination of Minimum Inhibitory Concentration (MIC)

The MIC of AgNPs for *P. aeruginosa* was determined by the broth microdilution method in 96-well plates, as recommended by the Clinical and Laboratory Standards Institute (CLSI, 2015) [72]. AgNPs were diluted in Mueller–Hinton broth (BD Difco, Franklin Lakes, NJ, USA) to concentrations ranging from 0,187 to 96,0 µg/mL. After incubation at 37 °C for 24 h, MIC values were defined as the lowest concentration of AgNPs capable of preventing visible microbial growth. Positive (*P. aeruginosa* incubated in the absence of AgNPs) and negative (AgNPs and broth only) controls were used.

#### 4.4.3. Time–Kill Assay

The time–kill assay followed the NCCLS standard protocols [73]. Bacterial strains were initially grown on Müller–Hinton agar (BD Difco, Franklin Lakes, NJ, USA) to 0.5 on the McFarland scale and then diluted 1:100 in Mueller–Hinton broth (BD Difco, Franklin Lakes, NJ, USA) to achieve a concentration of 1.5x10^6^ CFU/mL. The resulting solution was transferred to microtubes, at which each AgNPs were added at their respective MIC. The experiment was performed in triplicate, with incubation times of 1, 2, 3, and 4 h. At each time point, the aliquots were collected, diluted, and transferred onto MHA plates, which were then incubated at 37 °C for 24 h to determine the bacterial concentration in CFU/mL. The logarithmic curve for the CFU was constructed in the function of the time of incubation (time–kill). The graphic (Figure 7) was created using GraphPad Prism 9.1.1 (GraphPad Software Inc., Boston, MA, USA).

#### 4.4.4. Data Analysis Statistical

The data determination of minimum inhibitory concentration (MIC) was analyzed using the statistical package R, version 4.2.2 [74]. The results were subjected to the Shapiro–Wilk test of normality (*p* < 0.05), subsequently to the ANOVA test *(p* < 0.05), and the insignificant difference between the averages, the post hoc Tukey HSD test was applied (*p* < 0.05) in relation to the time–kill.

### 4.5. Cellulose Hydrogel Preparation

The nanocellulose used in this study was obtained as a 2% concentration paste (Exilva^®^, Borregaard VHV-S, Borregaard, Sarpsborg, Norway). This material was dispersed in purified water employing a rotor-stator (model Ultra Turrax T18, IKA^®^, Staufen, Baden-Württemberg, Germany) at 10,000 rpm for 10 min. After that, potassium sorbate was also dispersed in the gel to be used as a preservative (0.2 g/mL). The final concentration of the nanocellulose in the gel was 0.2 g/mL.

### 4.6. Hyaluronic Acid Hydrogel Preparation

A commercial 95% Sodium Hyaluronate (HA) low molecule weight grade was purchased in a 1% suspension component (Shandong Focuschem Biotech Co, Ltda, Jinning, Shandong, China). This ingredient was homogenized in purified water using a rotor-stator (model Ultra Turrax T18, IKA^®^ Staufen, Baden-Württemberg, Germany) at 10,000 rpm for 5 min until a solution structure was formed. The final concentration obtained was 0.01 g/mL.

### 4.7. Hydrogel Preparation of AH-Nanocellulose and AgNPs

Firstly, the nanocellulose and the hyaluronic acid solution were proportionally homogenized (1:1) utilizing a rotor-stator (model Ultra Turrax T18, IKA^®^ Staufen, Baden-Württemberg, Germany) at 10,000 rpm for 5 min, constituting the base release system (nano biocomponent). Subsequently, manual agitation was performed to incorporate 2 µL of the silver nanoparticle dispersion of the active principle in 30 g of the base gel.

### 4.8. Characterization of the Composite Consisting of Cellulose and Hyaluronic Acid

#### 4.8.1. Scanning Electron Microscopy (SEM)

A small portion of the sample was spread over a cover slip (a piece of around 1 cm²) and underwent a drying process in a room with temperature controlled at 20 °C for 48 h. The coverslip with the sample was fixed on a “stub” containing double-sided carbon tape. The sample received a layer of gold approximately 20 nm thick on the surface using the Sputter Coater equipment Bal-Tec, model SCD 050 (Bal-Tec Corp., Pfäffikon, Zürich, Switzerland), and was analyzed with an electronic microscope Philips, model: FEI Quanta 200 (Philips Co., Eindhoven, North Brabant, The Netherlands). The image analyses were made with ImageJ 1.54d software to measure the length and width of the fibrils cellulose.

#### 4.8.2. Fourier Transform Infrared Spectroscopy (FT-IR)

A gel sample in the form of dehydrated film was crushed and pressed using a hydraulic press. Potassium bromide pellets were formed, and transmittance readings were performed with a resolution of 2 cm^−1^ in the absorption range of 4000 to 400 cm^−1^. For analysis, a Bruker-Vertex 70 infrared spectrophotometer was used, equipped with an ATR accessory with a 45° Gcrystal.

#### 4.8.3. Differential Scanning Calorimetry (DSC)

The analyses were carried out in a Shimadzu DSC 60 calorimeter (Kyoto, Japan), in the temperature range of 25 to 450 °C, using hermetically sealed aluminum capsules, with a sample mass of approximately 2 mg, a heating ratio of 10 °C/min in a dynamic nitrogen atmosphere with a flow rate of 50 mL/min.

## 5. Conclusions

In conclusion, we developed a hydrogel composed of cellulose and HA incorporated with silver nanoparticles. AgNPs were synthesized based on green chemistry, using the amino acid arginine as a reducing agent and potassium hydroxide as a carrier. PDS measurements confirmed the average sizes of AgNPs at 57.88 nm, and microbiological analyses demonstrated an excellent antimicrobial action against strains of *Pseudomonas aeruginosa* (9027) at low concentrations of the agent, MIC for AgNPs was 1.5 µg/mL and without toxicity. The maximum action time of the hydrogel with AgNPs was 3 h. Therefore, the hydrogel developed is presented as an effective agent for the control and treatment caused by *P. aeruginosa.*

## 6. Patents

Registration number—BR1020220094497. Modesto, M.; Nakazato, G.; Kobayashi, R.K.T.; Andrade, G.J.S.; Souza, C.R.; Bigotto, B.G.; Celligoi, M.A.P.C.; Lonni, A.A.G.S.; Tischer, C.A. ‘Nanocellulose gel with hyaluronic acid and a silver nanoparticles with antimicrobial properties’, 2022. Category: Product. The institution where it was deposited: INPI—National Institute of Industrial Property. Country Brazil. Nature: Patent of Invention. Deposit date: 16 May 2022. Depositor/Holder: AINTEC—State University of Londrina-PR/Brazil.

## Figures and Tables

**Figure 1 antibiotics-12-00873-f001:**
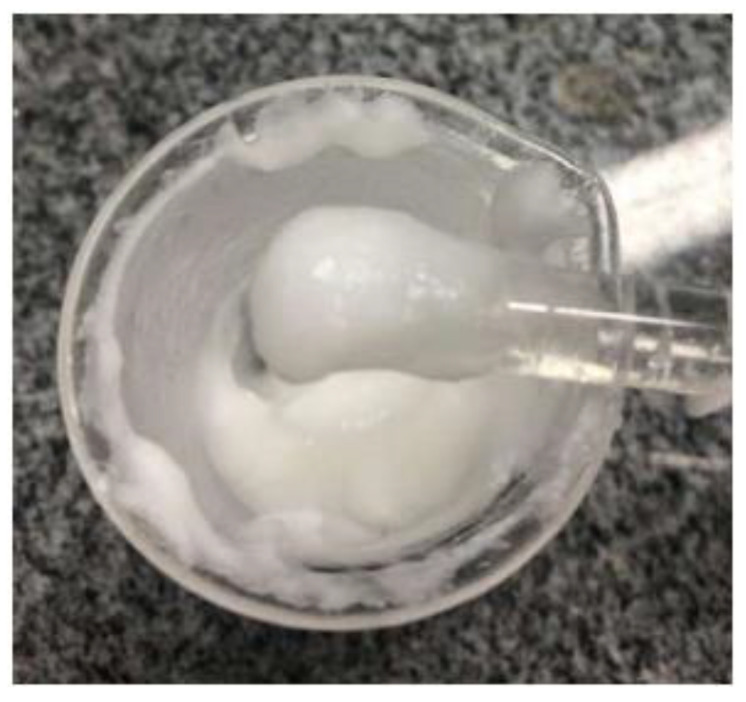
Visual characteristic of the hydrogel. The formulation is visibly white in color and exhibits a gelatinous appearance, indicating the full incorporation of all components of the formula.

**Figure 2 antibiotics-12-00873-f002:**
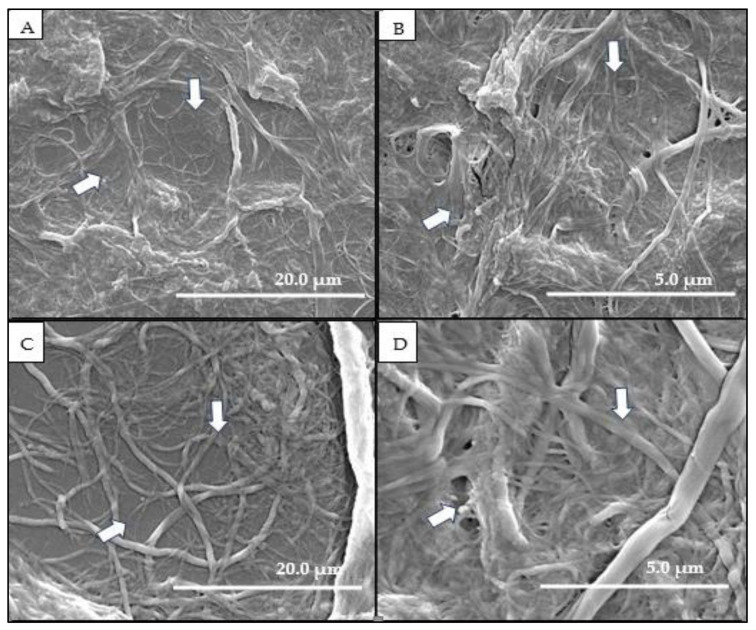
Scanning electron micrograph of cellulose gel and cellulose gel with HA. The images of the pure cellulose gel (**A**,**B**) randomly show the structure of the fibrils and empty spaces and pores (magnifications of 6000× and 24,000×), respectively. The images of the cellulose gel with HA (**C**,**D**) show the filling of empty spaces, the presence of pores, and thickening of the fibrils (6000× and 24,000×), respectively. Arrows: (**A**,**B**) empty spaces; (**C**,**D**) filling empty spaces, presence of pores, and thickening of fibrils.

**Figure 3 antibiotics-12-00873-f003:**
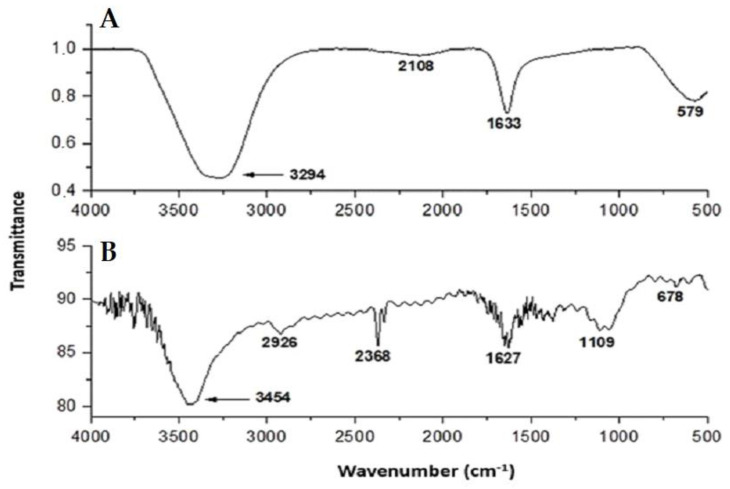
FT-IR spectrum. Pure cellulose (**A**) shows a characteristic cellulose band at approximately 3294 cm^−1^. Cellulose and HA (**B**) show a wider band exactly at 3454 cm^−1^, relating to the incorporation of the two biopolymers. The bands in (**A**) and (**B**) are attributed to the stretching of the bonds (OH and NH).

**Figure 4 antibiotics-12-00873-f004:**
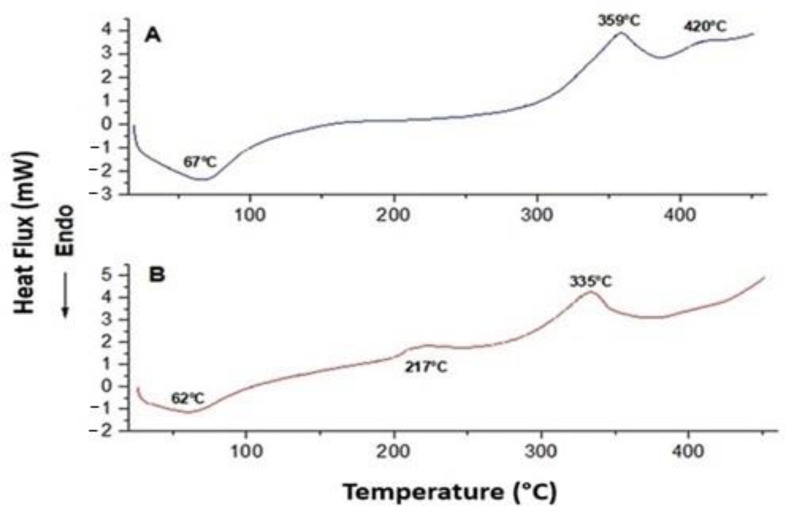
DSC curve (atmosphere N2, 50 mL/min). Pure cellulose (**A**) shows an endothermic event at 67 °C and two exothermic events at 359 °C and 420 °C. Cellulose with hyaluronic acid (**B**) shows an endothermic event at 62 °C and two exothermic events at 217 °C and 335 °C. A single mass loss process is observed in the composite formed between cellulose and HA, which indicates the interaction between the two biopolymers and their thermal stability.

**Figure 5 antibiotics-12-00873-f005:**
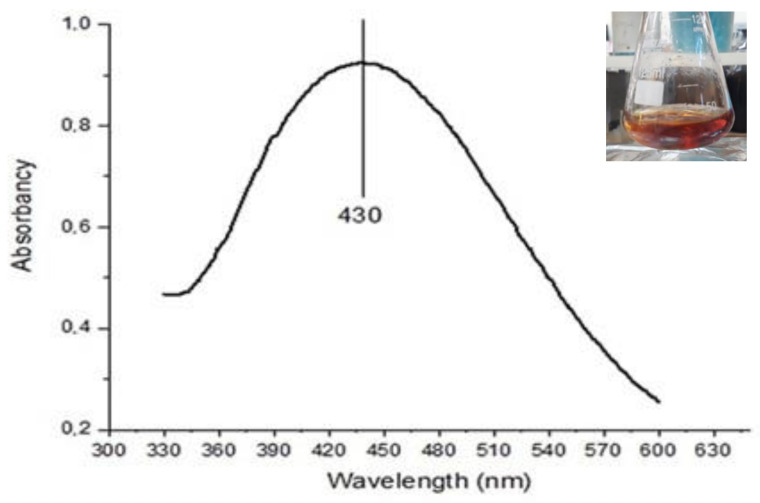
UV-vis absorption scan spectrum of AgNPs synthesized with arginine: KOH. The absorption band at 430 nm suggests the formation of nanoparticles. The inset shows the dark orange color of the stained solution.

**Figure 6 antibiotics-12-00873-f006:**
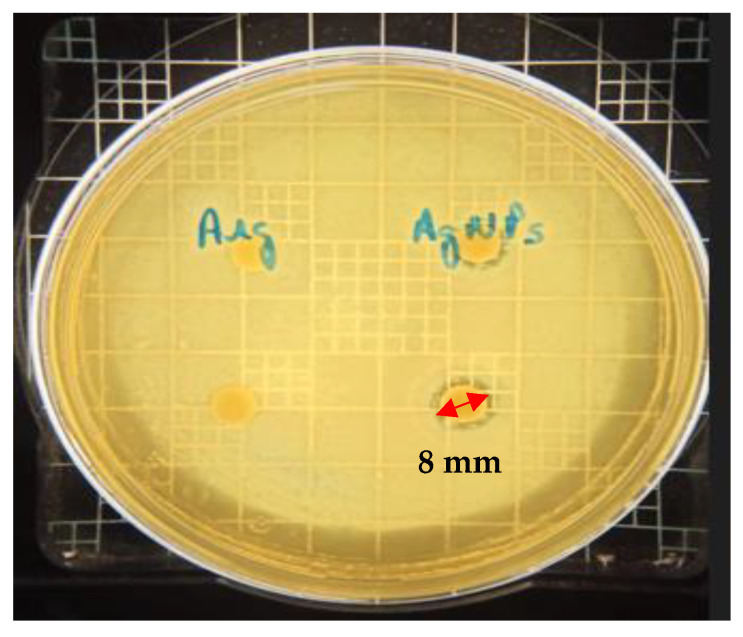
Disk diffusion on agar test performed with arginine (Arg) solution and silver nanoparticles (AgNPs) solution. The occurrence of an 8 mm diameter inhibition halo was observed in the AgNPs dispersion, proving its antibacterial effect.

**Figure 7 antibiotics-12-00873-f007:**
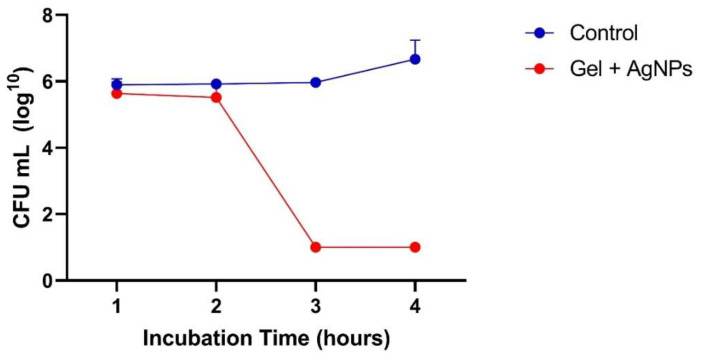
Time–kill curve of *P. aeruginosa* 9027 (ATCC). Bacteria at 5 × 10^5^ CFU/mL were exposed to treatment with the hydrogel containing AgNPs (red line). The control indicates bacterial growth without AgNPs (blue line).

**Figure 8 antibiotics-12-00873-f008:**
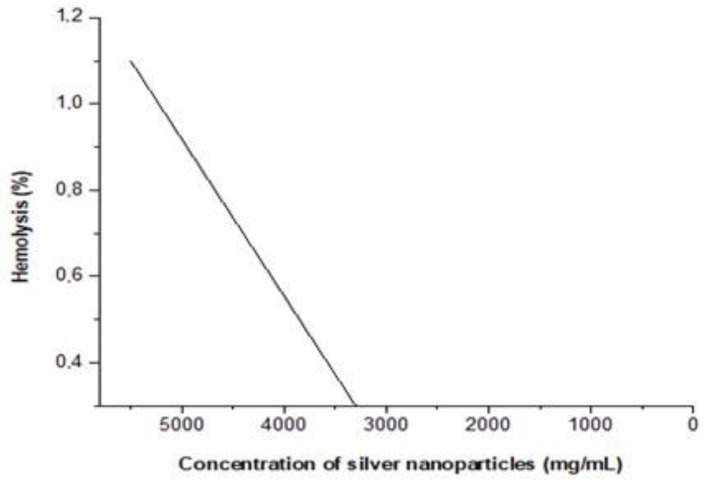
Hemolytic activity of different concentrations of AgNPs individually. Human erythrocytes exposed to AgNPs alone, ranging from 43.0 to 5500 mg/mL.

## Data Availability

Not applicable.

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
