# Peer review of "Cellulose Hydrogel with Hyaluronic Acid and Silver Nanoparticles: Sustained-Release Formulation with Antibacterial Properties against Pseudomonas aeruginosa"

_antibiotics, 2023, doi:10.3390/antibiotics12050873_

Round 1
Reviewer 1 Report
1- The introduction and discussion should be focused more on the observations and novelty of this study and supported with related references. The authors may use the following references
a. https://doi.org/10.3390/ijms24032191
c. Digest Journal of Nanomaterials and Biostructures 9 (2), 787-796
d. https://doi.org/10.3390/nano12162808
2. More concluding remarks must be also added.
3. No significant inhibition is observed, thus, biofilm assay is required
4. statistical analysis paragraph must be added to the materials and methods section.
5. Reagents with materials must be added with their origin (city, country).
Reviewer 2 Report
The article entitled “Cellulose hydrogel with hyaluronic acid and silver nanoparticles: sustained release formulation with antibacterial properties against Pseudomonas aeruginosa” submitted by Mirian Modesto is good and interesting work. This study describes the development of a topical hydrogel in a formulation composed of cellulose, hyaluronic acid, and silver nanoparticles against strains of Pseudomonas aeruginosa. However, after careful evaluation, it is revealed that the manuscript should be minorly revised. I, therefore strongly recommended the manuscript should be accepted after minor revision.
Comments:
1. Keywords should be replaced by eye-catching words.
2. In Fig 3, should use a scale bar. Also, write down the size of the nanofibers, also mention the length of nanofibers.
3. Author produces EDS or elemental analysis of the cellulose and composite.
4. FTIR spectra should comparatively discuss.
5. Apart from endothermic and exothermic temperature, the thermal stability should be discussed, at what temperature materials are thermally stable. If possible, find out the kinetics parameters of materials using degradation data.
6. Author stated, “The antibacterial activity of metallic compounds, mainly silver nanoparticles (Ag NPs), seems to be related to the release of silver atoms or ions [27,28], as well as the size and morphology of the nanoparticles (round, rod, triangle, etc.) [9,10,13, 29,30].” Authors are appealed to cite the following works for the statement.
Materials Today: Proceedings 29 (2020) 939-945; Materials Today: Proceedings 29(3) 720-725;
7. Authors are appealed to include killing kinetics mechanism pathogen with Ag NMs and composite. And provide sufficient control experiments (+ve and –ve) to validate the biological data collected. Also, compare all pathogens' results with standard antibiotics. How many times run the sample?
8. Authors are appealed to produce images of cytotoxicity assay with human red blood cells.
9. In conclusion, must produce an output of the present work. Highlight the novelty and findings of the work in the conclusion.
Reviewer 3 Report
The article by Modesto M. et al. investigates a complex dispersion of silver nanoparticles and nanocellulose in an aqueous solution of hyaluronic acid for use as an antibacterial gel. The authors provide data on the spreadability of the obtained dispersion, characterize it using SEM, DSC, FT-IR, UV-Vis, and DLS, and test it for antibacterial activity and cytotoxicity. The article is well written, though somewhat speculative. The authors call their system a hydrogel everywhere without providing clear evidence, such as a study of rheological properties. The article needs revision before publication.
Specific comments are as follows.
Line 25: It is necessary to replace "AgNPs solution" with "AgNPs dispersion", as silver nanoparticles form a dispersion rather than a solution.
Line 51: “The hydrogels are polymeric matrices formed by a porous three-dimensional network of crosslinked chains”. That's not an accurate statement. First, hydrogels can be from polymers that do not contain cross-linked chains, such as gels from gelatin or carrageenan, whose macromolecule conformation is helix-like. Second, hydrogels can be non-polymeric, such as hydrogels whose three-dimensional network consists of colloidal particles subject to coagulation. The following phrase is more appropriate: “The hydrogels are matrices formed by a porous three-dimensional network of crosslinked polymer chains, helix-like macromolecules or colloid particles”.
Line 91-94: “Therefore, the search for substitutes, such as more sustainable green chemistry, is the goal of many researchers [13,17]. Biologically active amino acids are environmentally friendly compounds suitable for green chemistry. Arginine presents…” Here one gets the impression that the authors were the first to suggest using amino acids as reducing agents for synthesizing silver nanoparticles and producing hydrogels, whereas this is not the case. Previously, an AgNPs-containing hydrogel was obtained using cysteine as a reducing agent (see 10.1039/c1sm06007d), which should be mentioned in the introduction.
Line 101: “nanocellulose”. Here the word "nanocellulose" appears for the first time, whereas earlier it was just "cellulose" hydrogels. The authors should mention nanocellulose closer to the beginning of the introduction (e.g., on line 53) rather than at the very end.
Line 111: “spreadability”. The authors should explain what this term means, and why there is a "high spreadability" in this case, i.e., what value is characteristic of poor, medium, or high spreadability. In other words, at the moment, Figure 2 does not contain any useful information and should either be deleted or discussed in more detail.
Line 203: “± standard deviation”. In Figure 8, the experimental points are shown without information about the standard deviation.
Line 365: “The particle size distribution”. The article did not provide data on the particle size distribution of silver nanoparticles. The authors might have given a histogram of the particle size distribution.
Line 418: “hyaluronic acid”. First, are the authors sure it was hyaluronic acid rather than sodium hyaluronate? Second, the authors should specify the molecular weight or intrinsic viscosity of the hyaluronic acid that they used.
Line 420: “until a gel structure was formed”. This is a speculative statement, as the authors do not provide a single proof of the gel structure in their samples. In addition, the authors contradict themselves since they write in the introduction that crosslinks are necessary for gel formation (line 51), whereas hyaluronic acid contains no cross-links and therefore forms a solution rather than a gel. The same is on line 423: “hyaluronic acid gels” -> “hyaluronic acid solution”.
Line 426: “silver nanoparticle solution” -> “silver nanoparticle dispersion”.
Round 2
Reviewer 1 Report
Accept
Author Response
Dear reviewer
Thank you for the acceptance. All comments and suggestions improved the quality of our manuscript.
Reviewer 3 Report
The authors have revised the manuscript appropriately. It can now be published.
Author Response

(The authors gave the same response as above.)
